

# Water activity and surface tension of aqueous ammonium sulfate and D-glucose aerosol nanoparticles

**Eugene F. Mikhailov[1], Sergey S. Vlasenko[1] and Alexei A. Kiselev[2]**

[1]Department of Atmospheric Physics, Saint Petersburg State University, St Petersburg, 199034, Russia
[2]Atmospheric Aerosol Research Department, Institute for Meteorology and Climate Research, Karlsruhe Institute of Technology (KIT), Hermann-von-Helmholtz Platz 1, 76344 Eggenstein-Leopoldshafen, Germany

*Correspondence to*: Eugene F. Mikhailov (eugene.mikhailov@spbu.ru)

**Abstract.** Water activity ($a_w$) and interfacial energy or surface tension ($\sigma$) are key thermodynamic parameters to describe the hygroscopic growth of atmospheric aerosol particles and their ability to serve as cloud condensation nuclei (CCN) influencing the hydrological cycle and climate. Due to size effects and complex mixing states, however, these parameters are not well constrained for nanoparticles composed of organic and inorganic compounds in aqueous solution.

In this study, we determined $a_w$ and $\sigma$ by differential Köhler analysis (DKA) of hygroscopic growth measurement data for aerosol particles smaller than 100 nm composed of aqueous ammonium sulfate (AS), D-glucose (Gl), and their mixtures. High-precision measurements of hygroscopic growth were performed at relative humidities (RH) ranging from 2.0% to 99.6 % with a high humidity tandem differential mobility analyzer (HHTDMA) in three complementary modes of operation: hydration, dehydration, and restructuring. The restructuring mode (hydration followed by dehydration) enabled the transformation of initially irregular particles into compact globules and the determination of mass equivalent diameters. The HHTDMA-derived growth factors complemented by the DKA, allows for determination of water activity and surface tension from dilute to highly supersaturated aqueous solutions that are not accessible with other methods. Thus, for mixed AS/Gl nanoparticles with mass ratio of 4:1 and 1:1, the upper limit of solute mass fraction ($X_s$) was 0.92 and 0.98, respectively.

For pure AS and Gl, the DKA-derived $a_w$ is in a good agreement with electrodynamic balance and bulk measurement data. For AS particles, our $a_w$ data also agree well with the Extended Aerosol Inorganic Model (E-AIM III) over the entire concentration range. In contrast, the UNIFAC model as a part of AIOMFAC was found to overestimate $a_w$ in aqueous Gl particles, which can be attributed to unaccounted intermolecular interactions.

For mixed AS and Gl nanoparticles, we observed a non-monotonic concentration dependence of the surface tension that does not follow the predictions by modelling approaches constructed for mixed inorganic/organic systems. Thus, for AS/Gl particles with a 1:1 mass ratio exhibited a strong decrease of $\sigma$ with increasing solute mass fraction, a minimum value of 56.5 mN m$^{-1}$ at $X_s \approx 0.5$, and a reverse trend of increasing $\sigma$ at higher concentrations. We suggest that D-glucose molecules surrounded by ammonium sulfate ions tend to associate, forming non-polar aggregates, which lower the surface tension at the air-droplet interface.

We analyzed the uncertainty in the DKA-derived water activity and surface tension, related to the instrumental errors as well as to the morphology of the nanoparticles and their phase state. Our studies have shown that under optimal modes of operation of HHTDMA for moderate aqueous concentrations, the uncertainty in $a_w$ and $\sigma$ does not exceed 0.2-0.4 % and 3-4 %, respectively, but it increases by an order of magnitude in the case of highly concentrated nanodroplet solution.

## 1. Introduction

The water uptake of aerosol particles is among the central issues of current research in atmospheric and climate (Pöschl, 2005; Andreae and Rosenfeld, 2008). Water activity and surface tension (interfacial energy) are key thermodynamic parameters of classical Köhler theory, which describes the hygroscopic growth of particles and their ability to serve as cloud condensation nuclei (CCN) (Swietlicki et al., 2008; Prisle et al., 2011; Forestieri et al., 2018; Ruehl et al., 2016; Ovadnevaite et al., 2017; Davies et al., 2019). Bulk methods, such as pendant drop tensiometry (Anastasiadis et al., 1987; Topping et al., 2007; Shapiro et al., 2009), Electrodynamic Balance (EDB) and optical tweezers operating with super-micrometric particles are mainly used to determine $a_w$ and $\sigma$ (Tang et al., 2019; Bzdek et al., 2020). The applicability of these methods to nanoparticles is limited because the water uptake, gas-particle and bulk-surface partitioning, as well as related phase transitions (deliquescence, efflorescence) depend on particle size (Biskos et al., 2006a,b; Prisle et al., 2010; Djikaev et al., 2010; Cheng et al., 2015). To bridge the gap between experimental and modelling results for bulk materials and



nanoparticles, a new method called Differential Köhler Analysis (DKA) was developed recently (Cheng et al., 2015). This method allows for determination of the water activity and surface tension of supersaturated aqueous solutions based on hygroscopic growth measurement of nano-sized droplets. Recently, Lei et al. (2023) used DKA method to estimate water activity of the levoglucosan and D-glucose aerosol nanoparticles using nano-HTDMA in the size range of 6-100 nm with RH below 90%. These results show a good agreement of the DKA-derived water activity with the earlier experimental data

and E-AIM model. However, no DKA-based surface tension data has been reported in that study.

Atmospheric aerosol particles are mainly a mixture of organic and inorganic components. The interaction of organic-inorganic species in aqueous solution affects both bulk (Raoul effect) and surface (Kelvin effect) properties of nanoparticles. In order to estimate these effects independently, model approximations for $a_w$ or $\sigma$ are introduced (Li and Lu, 2001; Tuckermann, 2007; Prisle et al., 2010; Petters and Petters, 2016; Ruehl et al., 2016; Ovadnevaite et al., 2017;

Forestieri et al., 2018; Davies et al., 2019; Schmedding and Zuend, 2023; and references therein). However, it is difficult to verify the validity of these approximations with respect to the nanodroplets accounting for the bulk-surface partitioning and liquid-liquid separation effects. The advantage of DKA method is precisely that it allows for simultaneous determination of the bulk and surface characteristics of nanodroplets in the wide concentration range including a highly supersaturated solution. This provides a new opportunity for validation of the theoretical approaches used for both $a_w$ and $\sigma$ in the pure

and mixed organic-inorganic aerosols. In addition to pure substances, the DKA method is applied here to derive the water activity and surface tension of two-component particles comprising ammonium sulfate and D-glucose with a mass ratio of 4:1 and 1:1. Both components are surface inactive and highly water soluble.

The DKA data were obtained from the measurements of hygroscopic growth factors with a high-humidity tandem differential mobility analyzer (HHTDMA) for aerosol particles with diameters in the range of 17-100 nm at relative

humidity (RH) of 2.0 - 99.6 %. The high precision of growth factor measurements in the wide particle size and RH range allows for determination of the water activity and surface tension from dilute to highly supersaturated aqueous solutions under conditions that are not accessible to other methods.

We analyze and discuss our nanoparticle measurement results with respect to those obtained with bulk experimental methods and thermodynamic models. In addition, our HHTDMA measurements provide information about

particle restructuring in response to water vapor adsorption at low and intermediate RH levels (Mikhailov et al., 2009, 2021), which enable the determination of mass equivalent diameters for the investigated nanoparticles as well as uncertainty analyses and refinements of the DKA-derived thermodynamic parameters at sub- and supersaturation conditions.

## 2. Material and methods

The investigated aerosols were produced by nebulizing an aqueous solution of pure ammonium sulfate (99.9% pure, ChemCruz) and D-glucose (99.55% pure, Fisher) at ~0.01% weight concentration or their mixture with AS:Gl = 4:1 and 1:1 weight ratio.

### 2.1 HHTDMA setup and modes of operation

The hygroscopic properties of size-selected aerosol particles were measured in the 2.0%–99.6% RH range with a high-humidity tandem differential mobility analyzer (HHTDMA) (see Mikhailov and Vlasenko, 2020; Mikhailov et al., 2021). Throughout the whole relative humidity range, the absolute uncertainty of RH is less than 0.5 %, and the relative growth factor uncertainty due to RH and instrumental errors does not exceed 1%. Three operation modes are available using this HHTDMA instrument: hydration and dehydration (H&D) also called restructuring mode, hydration, and dehydration. The

H&D mode was used to determine the optimal RH range in which initial irregular particles transform into compact globules.





Inorganic and organic aerosol particles as well as their mixtures restructure upon humidification below its deliquescence (Mikhailov et al., 2020). Irregular envelope shape and porosity cause a discrepancy between the mobility-equivalent and mass-equivalent particle diameters that limits precision of mobility-diameter-based hygroscopicity tandem differential mobility analyzer (HTDMA) (Gysel et al., 2004; Mikhailov et al., 2020). To account for restructuring, we used the minimum mobility particle diameter, $D_{b,H\&D,min}$, obtained in H&D HHTDMA mode as an approximation of mass-equivalent diameter of the dry solute particle, $D_s$ (i.e., $D_s = D_{b,H\&D,min}$). The size-dependent restructuring factor was calculated as follows: $g_{b,H\&D} = D_{b,RH}/D_{b,i}$, where $D_{b,i}$ and $D_{b,RH}$ is the initial mobility particle diameter and that measured at RH of aerosol pre-conditioning, respectively (Mikhailov et al., 2021, Suppl. S3).

In this study, the H&D mode was in situ coupled with a conventional hydration or dehydration mode. In the combined modes, the initial monodisperse dry particles were first led through the preconditioning section where they underwent microstructural transformation, by means of humidification in the 2%-90% RH range (NAFION humidifier) and drying (NAFION dryer followed by a silica gel diffusion dryer). At the same time the aerosol and sheath flow in the second differential mobility analyzer was below 3 % RH (Mikhailov et al., 2020; 2021, Suppl. S3). The preconditioning RH at which the particles reached their minimum diameter, $D_{b,H\&D;min}$ was then kept constant, throughout the standard hydration or dehydration experiment. The mobility-equivalent particle growth factor, $g_b$, was calculated as the ratio of the mobility-equivalent diameter, $D_b$, measured after conditioning (hydration, dehydration) to the minimum mobility diameter, $D_{b;H\&D;min}$, observed in the H&D mode:

$$g_b = \frac{D_b}{D_{b,H\&D,min}} \ . \tag{1}$$

### 3. Thermodynamic models

#### 3.1 Differential Köhler analysis (DKA)

The basis of the DKA method (Cheng et al., 2015) is the Köhler equation which describes the equilibrium relative humidity, $s_w$ over a spherical droplet as a function of $a_w$, σ and the mass equivalent diameter, $D_b$ :

$$s_w = \frac{RH}{100} = a_w \exp\left(\frac{4\sigma v_w}{RTD_b}\right) = a_w \exp\left(\frac{4\sigma v_w}{RTD_s g_b}\right) \ , \tag{2}$$

where $v_w$ is the partial molar volume of water, $R$ is the universal gas constant and $T$ is the temperature. $D_b$ in equation (2) is often replaced by the product of $D_s$ and $g_b$, that is, $g_b = D_b/D_s$. The logarithmic form of equation (2) is

$$ln s_w = ln\, a_w(g_b) + \frac{A}{D_s}\sigma(g_b) \ , \tag{3}$$

where $A = 4v_w/RTg_b$ . From Eq. (3) it follows that $a_w$ and σ can be obtained as the fitting parameters for the $s_w(D_s)$ dependence with the same $g_b$. Obviously, the larger the number of experimental values of $D_s$ that are used for the fitting, the more accurately $a_w$ and σ can be estimated. This study utilized four sizes for ammonium sulfate and 8 sizes for D-glucose and its mixture with ammonium sulfate in the $D_s$ range 17-100 nm. We used the Levenberg-Marquardt algorithm (implemented in Origin 9.0) to find $a_w$ and σ in the iterative procedure with standard fitting error as a measure of their uncertainty. It should be noted that fitting of the experimental $s_w(D_s)$ dependences over the entire set of dry diameters significantly improves the reliability of the DKA-derived $a_w$ and σ as compared to the original approach, where these quantities were calculated in pairs for available dry diameters (Cheng et al, 2015; Lei et al., 2023).





### 3.2 HHTDMA-derived activity coefficients

In a binary system at a constant temperature and pressure, the water activity, $a_w$ and the solute activity, $a_s$ are related by the
Gibbs–Duhem equation (Prausnitz et al., 1999):

$$x_s dln\, a_s = -x_w\, dln\, a_w \ , \tag{4}$$

where $x_w$ and $x_s$ are the mole fractions of water and solute, respectively. According to Robinson and Stokes (1970) for the
real solution the water activity and molal osmotic coefficient of solute, $\Phi_s$ are related by

$$ln a_w = -\frac{\nu_s m_s \Phi_s M_w}{1000} \ , \tag{5}$$

where $\nu_s$ and $m_s$ are the stoichiometric dissociation number and molality of the solute (mol kg$^{-1}$ of water), $M_w$ is the
molecular mass of water (g mol$^{-1}$). For electrolyte solution in the molality scale:

$$a_s = a_{\pm}^{\nu_s} = (\gamma_{\pm} m_{\pm})^{\nu_s} = \gamma_{\pm}^{\nu_s} m_s (\nu_+^{\nu_+} \nu_-^{\nu_-}). \tag{6}$$

$\nu_+$ and $\nu_-$ are the numbers of positive and negative ions produced upon dissociation of the solute ($\nu_s = \nu_+ + \nu_-$), $m_{\pm}, \gamma_{\pm}$
and $a_{\pm}$ are the mean molality, activity coefficient and activity of the solute. For ammonium sulfate with $\nu_s = 3$ and non-
dissociating D-glucose with $\nu_s = 1$ from Eq. (6) it follows:

$$a_{AS} = 4^{1/3} m_{AS} \gamma_{\pm}^3 \ \text{and} \ \ a_{Gl} = \gamma m_{Gl} \ . \tag{7}$$

Combining Eq. (4), Eq. (5) and Eq. (6) gives (Prausnitz et al., 1999):

$$dln\gamma_{\pm} = d\Phi_s + \frac{1}{m_s}\,(\Phi_s - 1)\,dm_s \ , \tag{8}$$

Integration from $m_s = 0$ to the solution of interest yields:

$$ln\gamma_{\pm} = \Phi_s - 1 + \int_0^m \frac{\Phi_s - 1}{m_s}\,dm_s \ . \tag{9}$$

The integral in Eq. (9) was evaluated numerically by plotting values of $(\Phi_s - 1)/m_s$ against $m_s$.

### 3.3 Surface tension models

The DKA-derived σ of pure AS and Gl aqueous solution and their mixtures were compared with the model of Li and Lu
(2001) and Dutcher et al., (2010). These models were moderately successful at representing the surface tensions of aqueous
solutions of some soluble salts and their mixtures with organic compounds (Topping et al., 2007). Dutcher et al., (2010)
model expresses the surface tension ($\sigma$) in terms of the surface tension of pure water ($\sigma_w$) and that of a hypothetical molten
salt ($\sigma_s$):

$$ln\sigma = x_w ln\,\sigma_w + x_s ln\,\sigma_s \ , \tag{10}$$

where $x_w$ and $x_s$ are the mole fractions of water ($w$) and salt ($s$). It is assumed that surface tension of solution at different
temperatures described by: $\sigma = \sigma_w + x_s F_{ws}(T)$ , with $F_{ws}(T) = a_{ws} + b_{ws}T$, where $a_{ws}$ and $b_{ws}$ are fitted parameters. At
very high salt concentrations, a similar relationship is used: $\sigma = \sigma_s + x_w F_{sw}(T)$ with $F_{sw}(T) = a_{sw} + b_{sw}T$. Two above
expressions can be applied to express the surface tension over the entire concentration range:




$$ln\sigma(T) = x_w\big(ln\ \sigma_w(T) + x_s F_{ws}(T)\big) + x_s\big(\ln(\ \sigma_s(T) + x_w F_{sw}(T)\big),\tag{11}$$

Equation (10) was used to calculate $\sigma$ of ammonium sulfate particles at T=298 K with parameters $\sigma_w$=72.0 mNm$^{-1}$; $\sigma_s$=185.0 mN m$^{-1}$; $a_{ws}=a_{sw} = 0$; $b_{ws}$= 0.366 mN m$^{-1}$ K$^{-1}$, $b_{sw}$= -0.289 mN m$^{-1}$ K$^{-1}$ as suggested by Dutcher et al. (2010). The Li and Lu (2001) surface tension model includes a Langmuir-type adsorption isotherm, and a term dependent on the activity of dissolved solid. For an aqueous solution with one solute, this model yields:

$$\sigma = \sigma_w + RT\Gamma_s^{w0}ln\frac{1}{1 + K_s a_s}\ ,\tag{12}$$

where $\Gamma_s^{w0}$ is the saturated surface excess of solute (s) and $K_s$ is the adsorption equilibrium constant of solute (s). For multi-component systems, Li and Lu (2001) proposed two approaches. The first approach, referred to as LiLu (1), implies that there is no interaction or competing adsorption between species at the interface:

$$\sigma = \sigma_w + RT\sum_{i-1}^{k}\Gamma_i^{w0}ln\frac{1}{1 + K_i a_i}\ ,\tag{13}$$

The second approach, that we refer to as LiLu (2) considers the interacting and competing adsorption between species in the interface for mixed aqueous systems at higher concentrations. This yields another expression for the surface tension of mixed solution:

$$\sigma = \sigma_w + RT\sum_{i-1}^{k}\Gamma_i^{w0}ln\left(1 - \frac{K_i a_i}{1 + \sum_i K_i a_i}\right)\ .\tag{14}$$

We also compared the DKA–derived $\sigma$ with simpler approximations, including those describing supersaturated solution. Thus, for ammonium sulfate solution, we used Pruppacher and Klett (1997) parameterization:

$$\sigma\ (N\ m^{-1}) = 0.072 + \frac{2.34 \cdot 10^{-2} \cdot X_s}{1 - X_s}\ ,\tag{15}$$

where $X_s$ is the mass fraction of solute. This parametrization is expected to be valid up to $X_s \approx 0.8$. For the concentration dependence of surface tension of an aqueous D-glucose solution the linear function given by (Aumann et al., 2010) was chosen:

$$\sigma(mN\ m^{-1}) = \sigma_w + \tfrac{\Delta\sigma}{\Delta C}C\ ,\tag{16}$$

where $C$ is the molarity concentration (mol l$^{-1}$) and $\Delta\sigma/\Delta C = 1.29$ (mN m$^{-1}$/mol l$^{-1}$) with upper limit of $C$ being is 3.5 mol/l ($m_{Gl} = 6.0$ mol kg$^{-1}$).

## 4 Experimental results and discussion

### 4.1 Particle restructuring due to humidification

Figure 1 illustrates the restructuring of aerosol particles in H&D mode. Both pure and mixed particles restructure with increasing RH, wherein this effect is strongest for largest diameters. Interestingly, mixed AS/Gl particles with a mass ratio of 4:1 undergo strong restructuring with $g_{b,H\&D}$ less than 1 (Fig.1c), whereas AS/Gl particles with a mass ratio of 1:1 show no structural change (Fig.1d). For these particles the restructuring factor, $g_{b,H\&D}$ larger than 1 and increases with RH, indicating that initial particles are compact and spherical (i.e., $D_{b,i} = D_{b,H\&D,min} = D_s$) (Mikhailov et al., 2020). The size-



170    dependent increase in $g_{b,H\&D}$ (Fig. 1c) is due to a residual water film on the particle surface after dehydration in H&D mode. The higher values of $dg_{b,H\&D}/dRH$ observed for particles with smaller diameters support this assumption. Since the H&D mode is in-situ coupled with a conventional hydration or dehydration mode the observed $D_{b,H\&D,min}$ values at specified RH were directly used to calculate the growth factor using Eq. (1). This has made it possible to eliminate the uncertainty in the growth factor caused by irregular particle morphology, which is as high as 5% or more for particles with $D_{b,i} > 70$ nm.

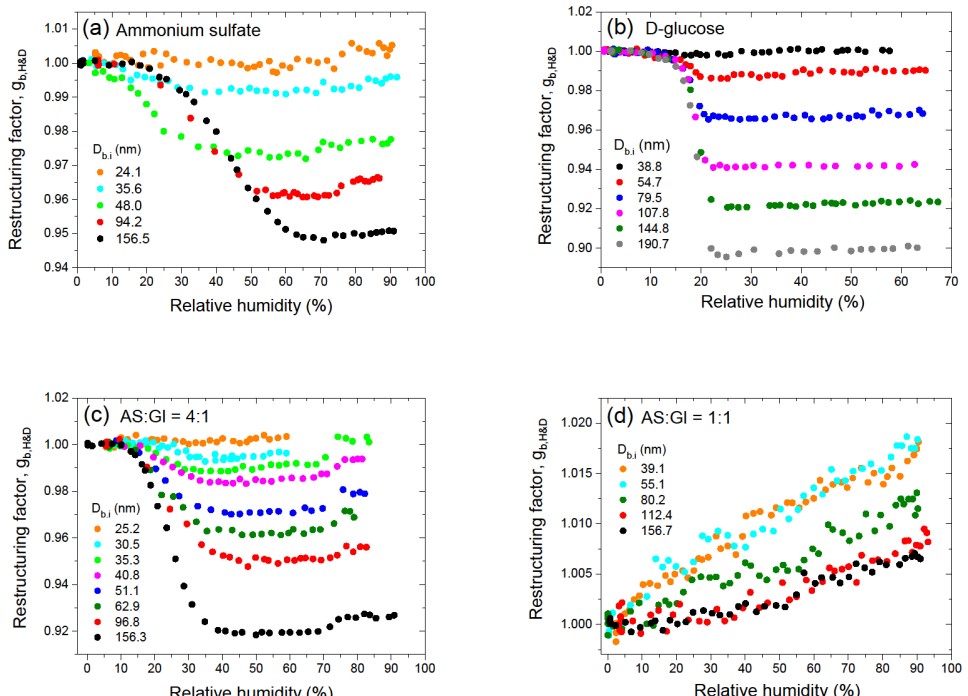

**Figure 1**. RH-dependent restructuring of ammonium sulfate (**a**), D-glucose (**b**) and their mixture with mass ratio AS:Gl = 4:1 (**c**) and AS:Gl = 1:1 (**d**) obtained in H&D mode as a function of particle size.

175

### 4.2 Size-dependent growth factors

Figure 2 shows the size-dependent growth factors of pure and mixed particles observed upon dehydration in the HHTDMA mode, calculated using Eq. (1). The dehydration branch was used because it allows us to determine $a_w$ and $\sigma$ over a wide concentration range, including highly concentrated droplet solutions. Note that, in contrast to pure ammonium sulfate and
180    its 4:1 mixture with D-glucose, in the case of pure D-glucose (Fig. 2b) and mixed AS/Gl particles with a mass ratio of 1:1 (Fig. 2d) the growth factors obtained in the hydration and dehydration modes are identical over the whole range of relative humidity (Fig.S1, Supplement). As expected, due to the Kelvin effect, for the same $g_b$ the equilibrium values of $s_w$ (or RH) for smaller particles shift towards higher values (Eq. 2). This is clearly visible for all particles measured at RH above 70% (inserts in Fig. 2). For ammonium sulfate particles, this pattern is valid also for highly concentrated solution up to
185    efflorescence RH (ERH ~ 30%) (Fig. 3a). However, in the cases of D-glucose (Fig. 3b) and AS/Gl (1:1) mixed particles (Fig. 3d) this pattern is not obvious. Moreover, in the case of AS/Gl (4:1) particles the values of $s_w$ for small particles are lower than for large particles (Fig.3c), in contrast to the behavior of pure ammonium sulfate (Fig. 3a). The measurement data in Fig. 3b and Fig. 3d also demonstrate that for D-glucose and AS/Gl (1:1) particles the growth factor is not a strictly




monotonic function of RH. The inserts in Fig. 3b and Fig. 3c show that at RH about 48 % the derivative $dg_b/dRH$ has an
inflection point, indicating that phase state of the particles before and after this point is different. Most likely at low RH D-
glucose and AS/Gl (1:1) mixed particles are in the semi-solid amorphous state, the increase of humidity over glassy
amorphous particles leads to a moisture-induced phase transition that occurs at "glass transition relative humidity" of $RH_g \approx$
48% (Mikhailov et al., 2009).

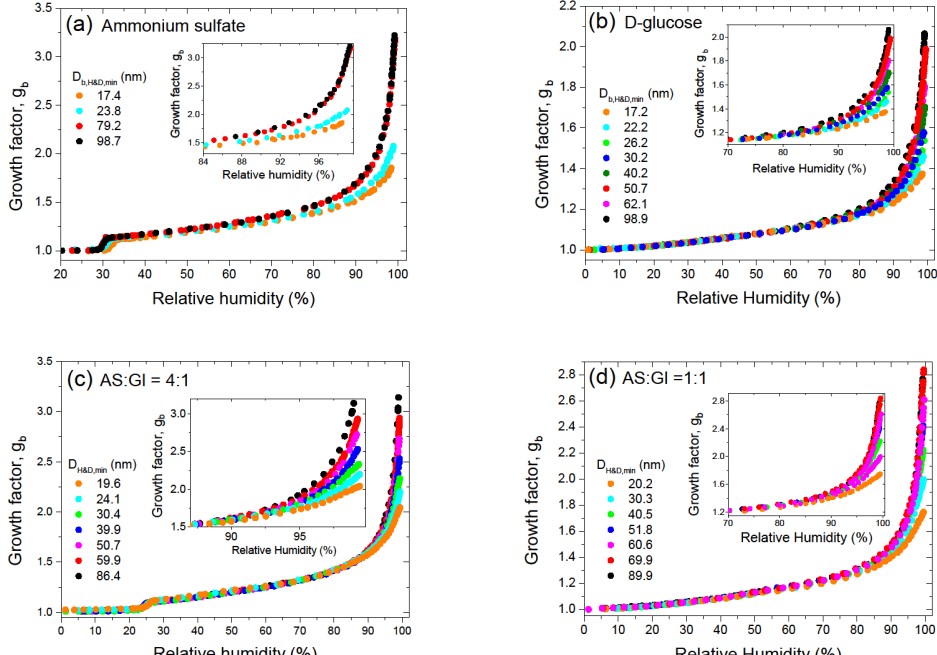

**Figure 2**. Size-dependent growth factors, $g_b$ of ammonium sulfate (**a**), D-glucose (**b**) and their mixture with mass ratio
AS:Gl = 4:1 (**c**) and AS:Gl = 1:1 (**d**) obtained in the dehydration HHTDMA mode.

Continued water uptake converts the amorphous particles into concentrated solution droplets. Due to the very low molecular
diffusivity of glasses, the uptake of water vapor by glassy aerosol particles is limited to surface adsorption, whereas above
$RH_g$ the particles are in a viscous liquid state and absorb water in the particle bulk. Differences in water sorption
mechanisms may explain why the growth factors of small particles at low RH are higher than for larger particles. At RH
bellow $RH_g$ with the same water adsorption layer, the water film contribution to $g_b$ will be higher for small particles. For
example, at a water layer thickness of 0.56 nm (i.e. 2 monolayers of water molecules), $g_b$ for aerosol particles of 19 and
100 nm are 1.031 and 1.006, respectively. This pattern is clearly seen for AC/Gl (1:1) particles with compact initial structure
(i.e. $D_{b,i} = D_{H\&D,min}$ and $g_b = g_{H\&D}$), for which at the same RH the growth factor of large particles is lower than that for
small particles (Fig.1d). Note also that moisture-induced transformation from semi-solid state to solute state can occur
gradually within a certain RH range (Mikhailov et al., 2009). All these features in the water uptake by amorphous particles
at high concentrations (low growth factor values) lead to additional uncertainties when using the DKA method.



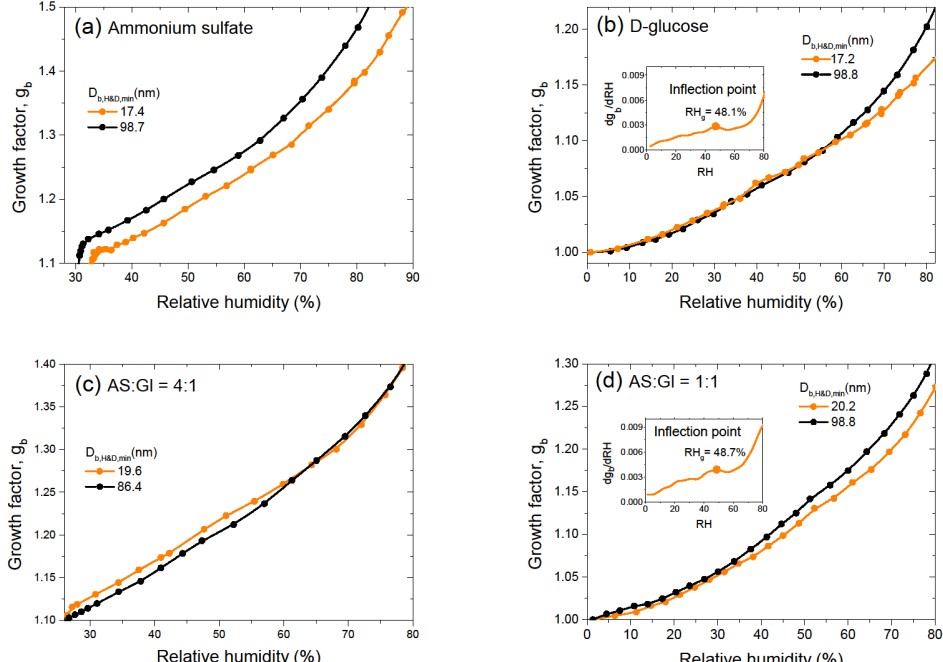

**Figure 3**. Growth factor, $g_b$ of ammonium sulfate (**a**), D-glucose (**b**) and their mixture with mass ratio AS:Gl = 4:1 (**c**) and AS:Gl = 1:1 (**d**) at low RH values for minimum and maximum sizes. The Inserts (**b, d**) show derivative, $dg_b/dRH$ with inflection point, $RH_g$. The instrumental error in $RH_g$ does not exceed the symbol size.

### 4.3 DKA-derived $a_w$ and $a_s$ for single component aqueous solution

The DKA-derived water activity for ammonium sulfate and D-glucose aerosol particles are shown in Fig. 4a and Fig.4b, respectively. More information on the DKA calculation can be found in Supplementary S1. The estimated uncertainty of $a_w$ for pure species is shown in Fig. S2 (Supplement). The $a_w$ values for ammonium sulfate (Fig. 4a) agree well with

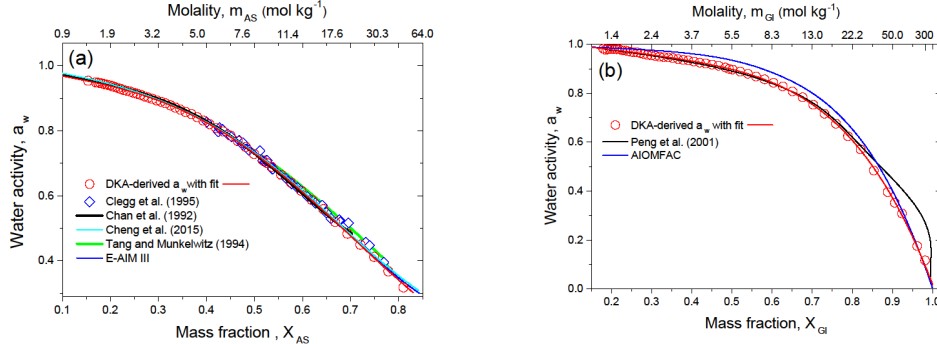

**Figure 4.** DKA-derived water activity, $a_w$ of ammonium sulfate (**a**) and D-glucose (**b**) aqueous solution at 298 K in comparison with literature data as a function of mass fraction and molality. The red line in (**a**) and (**b**) is a polynomial fit to all DKA data of $a_w$. The best fit coefficients are listed in Table S1.




electrodynamic balance (EDB) measurement of Tang and Munkelwitz (1994), Clegg et al. (1995), Chan et al. (1992) and are almost identical to the DKA-derived $a_w$ previously reported by Cheng et al. (2015) and Extended Aerosol Inorganic Model (E-AIM III) of Wexler and Clegg (2002), including highly concentrated aqueous solutions. In general, the DKA-derived water activities obtained for pure ammonium sulfate particles reproduce well literature data.

Figure 4b shows the DKA-derived $a_w$ of D-glucose aerosol particles. Our data are consistent with the EDB measurements of Peng et al. (2001) up to $X_{Gl} \sim 0.8$. The observed divergence of the water activity at $X_{Gl} > 0.8$ may be due to kinetic limitations in highly viscous semi-solid amorphous particles (Mikhailov et al., 2009) which prevail in EDB experiment with micron-sized particles. As noted in Sect. 4.2 and shown in Fig. 3d at $RH = RH_g \sim 48\%$ particles undergo moisture–induced phase transition where semi-solid particles become concentrated solution droplets. This RH value is very

close to $a_w \sim 49$ %, below which strong deviation between DKA- and EDB-derived $a_w$ is observed (Fig.4b). If this is the case, the DKA-derived $a_w$ here is closer to the real $a_w$.

At the same time, the DKA-derived $a_w$ differ from those predicted by the AIOMFAC model (Zuend et al., 2008) with maximum deviation of  6 % at $X_{Gl} \sim 0.7$.  The discrepancy between measured and AIOMFAC predicted $a_w$ values for carbohydrates has been noted previously and is attributed to relatively strong intramolecular interactions due to several

polar groups in close proximity, not accounted for by the UNIFAC model within the AIOMFAC framework (Zuend et al. 2011).

The multicomponent surface tension model of Li and Lu (2001) is based on the activity of the solute.  To take advantage of this model, we calculated the activity coefficients of single solutes by numerical integration of Eq. (9) and then, using Eq. (7), their activities (see Supplement, Sect. S2 for more details). The calculation results are shown in Fig. 5

in comparison with literature data. One can see that the ammonium sulfate activity is in a good agreement with E-AIM model (Fig. 5a) up to $X_{AS} = 0.75$ ($m_{AS} = 22.7$ mol kg$^{-1}$). At higher $X_{AS}$ values, there is a discrepancy between our data and the model data. This discrepancy is most likely due to the uncertainty in DKA-derived $a_w$ in concentrated ammonium sulfate aqueous solution Fig. S2 (Supplement) and uncertainty in assessment of the integral Eq. (9) in the asymptotic region ($m_s \to 0$) (Lakhanpal and Conway, 1960).

Figure 5b shows the DKA-based activity of D-glucose.  One can see that these values are in agreement with bulk measurement of Miyajima et al. (1983), obtained by the isopiestic method (blue symbols) available up to $X_{Gl}=0.52$ ($m_{Gl}=6.0$ mol kg$^{-1}$). At the same time, a systematic deviation is observed between our results and the AIOMFAC model. As previously noted, the UNIFAC as a part of AIOMFAC model does not match well with D-glucose and other carbohydrates owing to strong intramolecular interactions of the polar groups in close proximity.

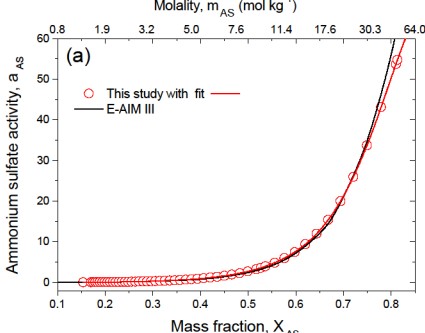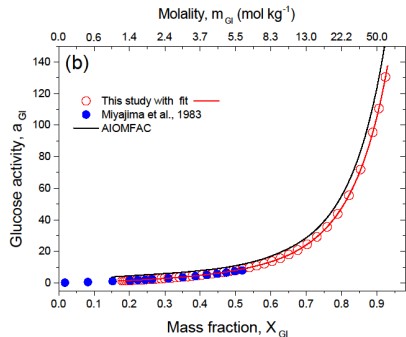

**Figure 5.** Solute activity of ammonium sulfate (**a**) and D-glucose (**b**) in water at 298 K together with literature data as a function of mass fraction and molality. The red line in (**a**) and (**b**) are polynomial function with fitting parameters listed in Table S2.




In general, the DKA-derived water activity and activity of ammonium sulfate and D-glucose in aqueous solutions reproduce well the literature data. For future applications, we fitted DKA-derived $a_w$ and $a_s$ for single solutes with a polynomial function. The obtained fitting parameters are listed in Table S1 and Table S2, respectively (see Supplement).

**4.4 Surface tension of the single solute solution droplet**

Figure 6 shows the DKA-derived surface tension of ammonium sulfate and D-glucose aqueous solution droplets as a
function of solute activity. For future use, we fit our data with Li and Lu (2001) model (Eq.12). The resulting values of $\Gamma_s^{w0}$ and $K_s$ together with the $a_s$ fitting interval are listed in Table S3. The negative $\Gamma_s^{w0}$ values obtained for both ammonium sulfate and glucose solution droplets indicate that a negative adsorption occurs on the interface between the vapor and liquid phases leading to surface tensions increase with concentration, which is typical for surface-inactive compounds (Tuckermann, 2007; Aumann et al., 2010).

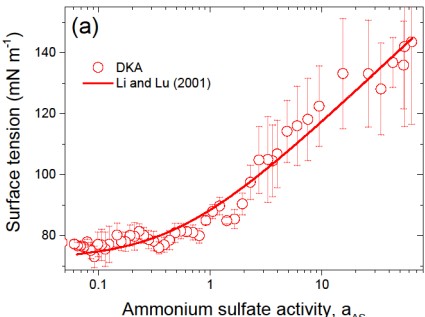
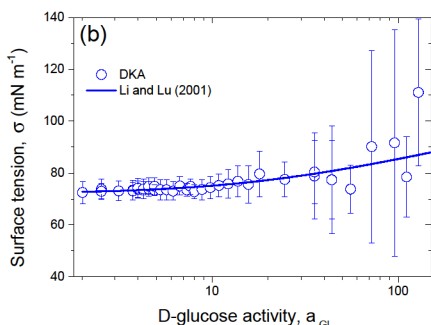

**Figure 6**   Surface tension of ammonium sulfate (**a**) and D-glucose (**b**) aqueous solution droplets as a function of solute activity. The line in (**a**) and (**b**) is the model of Li and Lu (2001) (Eq. 12) with the best-fit parameters listed in Table S3.


For comparison with the literature data, in Fig. 7 we plotted the DKA surface tension in a more accessible concentration scales, i.e. in mass fraction and molality. Figure 7a shows that DKA-derived σ for ammonium sulfate solution is in agreement, within uncertainty, with the empirical approximation of Pruppacher and Klett (1997) and also consistent with

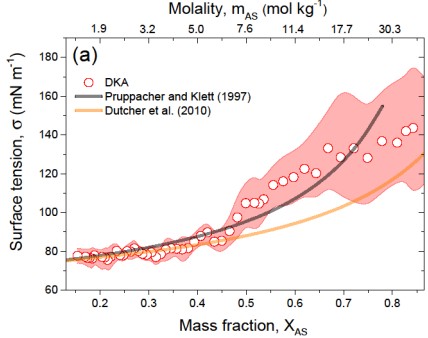
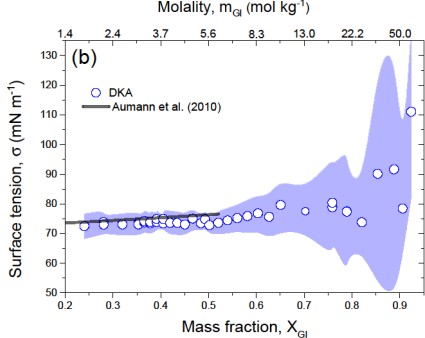

**Figure 7.** DKA-derived surface tension of ammonium sulfate (**a**) and D-glucose (**b**) solution droplets as a function of mass fraction and molality in comparison with literature.  The shaded area in (**a**) and (**b**) indicates uncertainty in the DKA retrieval σ.





Dutcher et al., (2010) model up to $X_{AS} \approx 0.5$ ($m_{AS} = 7.6$ mol kg$^{-1}$). Above this value, the difference between DKA data and those of Dutcher et al. (2010) gradually increases with increasing concentration. This difference is due to the model parameters used in Dutcher et al. (2010) (Sect. 3.3) are only valid for a concentration of ammonium sulfate not exceeding $X_{AS} = 0.43$ ($m_{AS} = 5.69$ mol kg$^{-1}$), although the model structure allows for calculation of $\sigma$ in the concentration range from dilute solution to molten salt (Eq.11). Figures 7b shows the DKA-derived $\sigma$ for D-glucose solution droplets as a function

of mass fraction or molality together with the literature data. It can be seen that the DKA data are consistent with bulk measurements reported by Aumann et al. (2010). In the $X_{Gl}$ range of 0.24 to 0.52, the average deviation between DKA-derived and literature $\sigma$ is of 2%.

Note that the uncertainty in the DKA-retrieval parameters is strongly influenced by of the accuracy of $s_w$ (or RH) and $g_b$ determination. At low $g_b$ (highly concentrated solutions) when $\Delta g_b/\Delta s_w$ is small, the uncertainty in the DKA-

derived $a_w$ and $\sigma$ increases with decreasing $g_b$. In addition, phase state ambiguity and associated kinetic limitations can provide additional uncertainties. As an example, Fig.S3 (Supplement) shows experimental $s_w(D_s)$ dependences at low and high growth factors for D-glucose aerosol particles and corresponding uncertainties in the DKA-derived $a_w$ and $\sigma$.

**4.5 DKA-derived $a_w$ and $\sigma$ for mixed ammonium-sulfate and D-glucose aerosol particles**

In the absence of solute-solute interactions ("separate solutes"), the corresponding relationship for the water activity of the

mixture is: $a_w = \prod_i a_{w,i}$ , where $a_{w,i}$ is the water activity of a pure aqueous solution of $i$ at the same concentration as in the mixture (Mikhailov et al., 2004; Clegg et al., 2006). Within this approach, referred to as "separate solute water activity" (SSWA), the water activity of mixed particles containing ammonium sulfate and D-glucose can be expressed as:

$$a_w = a_{w,AS}a_{w,Gl} \tag{17}$$

Figure 8 shows the DKA-derived water activity of the mixed AS/Gl particles with a mass ratio of 4:1(**a**) and 1:1 (**b**), and

results of the AIOMFAC and SSWA models. One can see that the DKA-derived $a_w$ and both models are consistent up to $X_s \sim 0.3$. A further increase in concentration is accompanied by a significant deviation of the model $a_w$ values, both among themselves and from the DKA-derived $a_w$. It appears that the SSWA approximation underestimates the specific

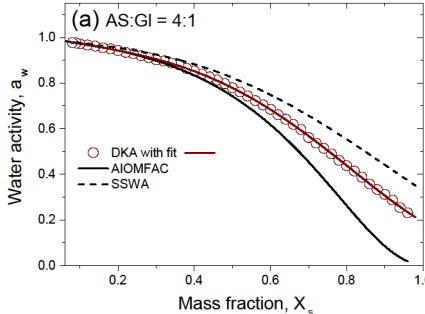 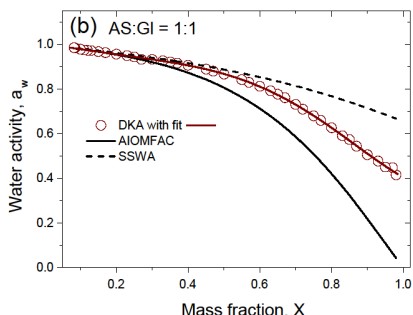

**Figure 8.** Water activity of the aqueous solution of the mixed ammonium sulfate and D-glucose with mass ratio AS:Gl = 4:1 (**a**) and AS:Gl = 1:1 (**b**) compared to model data as a function of the mass fraction of the solute, $X_s$ . The red line in (**a**) and (**b**) are polynomial function with fitting parameters listed in Table S1. SSWA denotes separate solute water activity (Eq.17)



interactions between ions and molecules in the droplet volume (higher $a_w$), whereas the AIOMFAC model overestimates

them (lower $a_w$), especially in concentrated solutions.

Figure 9 shows the DKA-retrieved surface tension of the mixed AS/Gl particles with a mass ratio of 4:1 (**a**) and

1:1 (**b**), compared to Li and Lu (2001) model. Both approaches, i.e. LiLu (1) and LiLu (2) do not match well the HHTDMA-

DKA-based σ. This discrepancy is especially pronounced for the mixed particles with an AS/Gl mass ratio of 1:1 (Fig.9b).

A slight lowering of the surface tension compared to that of water is already noticeable for a 4:1 mixture (Fig.9a). The

AS:Gl = 1:1 mixture (Fig.9b) shows more significant surface tension depression, which is not a monotonic function of

concentration. At $X_s \approx 0.5$ the σ reaches a minimum of 56.5±3.0 mN m⁻¹ (22 % reduction compared to pure water) and

increases again with concentration. Further increase in surface tension with concentration is the result of solidification of

aerosol particles, i.e. their transition from liquid to glassy state. Note that the observed pattern is reproduced in the repeated

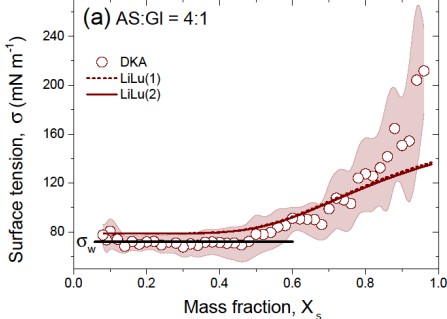
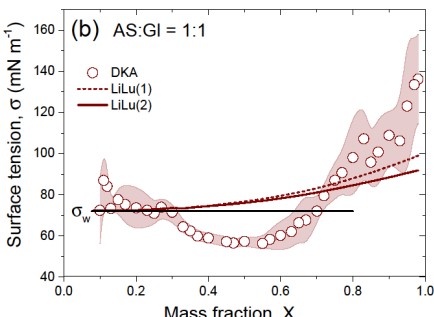

**Figure 9.** DKA-derived surface tension of the aqueous solution of the mixed ammonium sulfate and D-glucose with mass
ratio AS:Gl = 4:1 (**a**) and AS:Gl = 1:1(**b**) together with Li and Lu (2001) model (LiLu (1) - Eq. (13) and LiLu (2) - Eq. (14))
as a function of mass fraction of the solute, $X_s$. The shaded area in (**a**) and (**b**) indicates the uncertainty in the DKA retrieval
σ , the black line is the surface tension of pure water σ_w = 72.0 mN m⁻¹.

measurements (see Fig.S4, Supplement). Most likely the reduction in surface tension at moderate concentrations is caused

by salting out effect (Setschenow, 1889; Kiss et al., 2005; Marcolli and Krieger, 2006; Frosch et al., 2011; Wang et al.,

2014; Lin et al., 2020; Bzdek et al., 2020 ) and results from the interaction between ammonium sulfate ions and D-glucose

molecules, facilitating the association of initially surface-inactive hydrophilic organic molecules into surface-active

hydrophobic associates (quasi-macromolecules). Docoslis et al. (2000) analyzed the difference between polysaccharides

(dextran, ficoll) and their surface-inactive constituents (sucrose, glucose). They concluded that the contrasting results are

caused by the differences in polar intermolecular reactivities of the monomeric and polymeric glucides. The singly dissolved

monomeric sugars contribute strongly to the polar (Lewis acid-base) free energy of cohesion through the multiple

interactions between their freely available electron donors and acceptors, through mutual interactions as well as through

interactions with the surrounding water dipoles. However once covalently polymerized, these polysaccharides have lost the

strong bipolarity of the monomeric sugar molecules. It can thus be assumed, that in an aqueous AS/Gl system NH₄⁺ and

SO₄²⁻ ions effectively neutralize the bipolarity of the monomeric D-glucose molecules, facilitating their association into less

polar aggregates with reduced σ values, which are more readily accommodated at the air-droplet interface. Obviously, this

mechanism is more pronounced for mixed AS:Gl = 1:1 particles (Fig.9b) than for their 4:1 mixture (Fig.9a), due to the

higher D-glucose content (by a factor of 2.5).

**5 Summary and conclusions**



In this study, the DKA method was applied to derive water activities and surface tension of pure ammonium sulfate and D-glucose and their mixtures with mass ratios of 4:1 and 1:1 based on shape-corrected hygroscopic growth factors for particles with diameter of 17-100 nm in the relative humidity range of 2.0 - 99.6 %. The obtained $a_w$ and σ for pure ammonium sulfate and D-glucose droplet solution are in a good agreement with the bulk measurements in the available concentration range for bulk methods. The DKA method was employed for the first to determine $a_w$ and σ of ammonium AS/Gl mixed particles. Our data show that for dilute and moderate solution concentrations ($X_s$ <0.3) the SSWA and AIOMFAC models are in agreement with DKA-derived $a_w$. However, the discrepancy between the model and our data increases rapidly with increasing solution concentration, with SSWA underestimating DKA data, while AIOMFAC overestimates it. Both ammonium sulfate and D-glucose are surface-inactive compounds with a positive Δσ/ΔC slope indicating negative solute adsorption. Interestingly, mixing them together leads to positive adsorption and reduction in surface tension at the air-surface interface, i.e., due to salting out the AS/Gl mixture become surface-active compound. We suggest that AS ions neutralize polar groups of D-glucose, helping them to combine into less polar aggregates with reduced σ values at the air-surface interface.

The error analysis showed that there are some factors affecting the accuracy of $a_w$ and σ determined by the DKA method. One of them is the irregular morphology of the initial nanoparticles, which leads to a size-dependent error in $g_b$. Other factors are mainly characteristic of highly concentrated nanodroplet solutions, such as the large uncertainty of $g_b$ and ambiguity of the phase state. Both small growth factor changes at low RH and the different mechanisms of water uptake (surface adsorption vs. bulk absorption) lead to an increased uncertainty in the DKA-derived values of $a_w$ and σ. Thus, for D-glucose aerosol particles at $RH$ near $RH_g$= 48 % and $g_b$=1.08 ($X_{Gl}$ = 0.85; $m_{Gl}$=32.4 mol kg$^{-1}$) the relative uncertainty in $a_w$ is 3.5% and in σ is 41.2 %, respectively. Regardless of considered limitations, HHTDMA-DKA remains the only method for obtaining thermodynamic parameters for highly concentrated nanodroplet solutions.

In general, our results show that combination of high precision HHTDMA-based growth factor measurements with DKA approach allows for determination and analyses of water activity and surface tension of pure and mixed aerosol particles of dilute and highly supersaturated aqueous solutions under conditions that are not accessible to other methods. The results obtained in this study can be helpful for refining the interaction parameters of D-glucose and its mixture with ammonium sulfate in thermodynamic models such as AIOMFAC and E-AIM.

Additional studies are needed to better understand interaction between organic and inorganic components and their effects on the water activity and surface tension. In future work we plan to apply the DKA method to mixed organic/inorganic nanoparticles containing surface-active organic species.

**Data availability**. Raw data used in this study are archived and are available on request by contacting the corresponding author.

**Supplement.** The supplement related to this article is available online at:

**Acknowledgements.** We would like to thank Ulrich Pöschl, Yafang Cheng and Hang Su for their helpful recommendations that improved this manuscript.

**Author contributions**. EFM designed the study, performed the concomitant measurements, carried out the data analysis, and wrote the manuscript with input from all coauthors. SSV and AAK contributed to the discussion of the results.

**Competing interests.** Authors declare that they have no competing interests.



**Financial support.** This work was supported by the Russian Science Foundation, project 22-27-00258.

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
