# Peer review of "Water activity and surface tension of aqueous ammonium sulfate and D-glucose aerosol nanoparticles"

_EGUsphere, 2023_

## Referee Comment (RC2)

*Comments:*

This work reported water activity and surface tension of aqueous AS and D-glucose aerosol particles using DKA. The manuscript fits well to the scope of ACP. I recommend it to be published after the following comments have been adequately addressed.

1. I am worried about the novelty. AS and D-glucose chemicals are not new in hygroscopicity study. Growth factor, water activity and surface tension have been reported in many papers, but not cited in the manuscript. I would suggest the authors demonstrate the new findings (maybe the mixtures and high RH) relative to the previous studies.

2. Section 4.1: why is the distinct trend of restructuring of AS/Gl particles ($g_d$) between mass ratio of 4:1 and 1:1, as shown in Figs.1c and 1d.

3. Line 186: any explanation?

4. Line 190: is there any evidence about the phase state description?

5. Figure 9b: could you explain why the surface tension decrease firstly, and then increase along with increasing solution concentration?

6. Is it possible to provide the parametrization for water activity and surface tension of different chemicals using DKA? Then the method could be used more widely.

---

## Author Comment (AC1)

Response to referee #1

The referee's comments are in italics, our responses in plain font.

*In this manuscript, the authors derived water activity and surface tension from hygroscopic growth of aerosol particles smaller than 100 nm composed of aqueous ammonium sulfate, D-glucose, and their mixtures. The hygroscopic growth of particles was measured over a wide range of RH from 2.0% to 99.6% using a high humidity tandem differential mobility analyzer (HHTDMA). The derived water activity and surface tension were compared with those from electrodynamic balance and bulk measurements and thermodynamic model predictions (E-AIM and UNIFAC). Overall, the manuscript is well written, and the topic fits the scope of Atmospheric Chemistry and Physics very nicely. I recommend the manuscript for publication after the authors address the following comments.*

We thank referee #1 for suggestions for improvement that were taken into account upon manuscript revision. Responses to individual comments are given below.

*Major comment:*
*The authors derived water activity and surface tension using differential Kohler analysis, i.e., fitting parameters for the $s(D_s)$ dependence with the same $g_b$ (i.e., Eq. 3). This approach assumes that water activity and surface tension depend on $g_b$ only (i.e., independent of $D_s$). However, due to surface-bulk partitioning, both water activity and surface tension also depend on $D_s$ (i.e., surface area to volume ratio). I would suggest that the authors include relevant discussions on how such dependence affects the accuracy of derived water activity and surface tension and conclusion of this study.*

Line 122 and below, the following text has been added:
According to Eq.(3), DKA assumes that water activity and surface tension depend on $g_b$ only (i.e. concentration). However, in case of bulk-surface partitioning, both water activity and surface tension may also depend on $D_s$. To minimize this effect, we used surface inactive compounds. Previous studies on the example of NaCl and $(NH_4)_2SO_4$ nanoparticles have shown that the size effect is negligible for such compounds (Bahadur and Russell, 2008; Cheng et al., 2015, Supplement)

New reference: Bahadur, R. and Russell, L. M.: Effect of surface tension from MD simulations on size-dependent deliquescence of NaCl nanoparticles, Aerosol Sci. Technol.., 42(5), 369-376, https://doi.org/10.1080/02786820802104965, 2008.

Line 326 and below, new text has been inserted instead of old one:
As mentioned above, DKA assumes that thermodynamic parameters depend only on concentration and are independent of particle size. Good agreement of DKA-derived $a_w$ and σ with the literature data, showed that for particles in the size range of 20-100 nm, this approach is acceptable for single-component solutions of AS and Gl and their mixtures, at least at moderate concentrations. Additional HHTDMA-DKA studies are needed to evaluate the accuracy of this approximation for nanoparticles containing surface-active molecules. We plan to conduct such studies to compare the DKA-derived $a_w$ and σ with available experimental data and models that account for size-dependent bulk-surface partitioning.

*Minor comments:*

*Line 188-189: The growth factor is a monotonic function of RH (first derivative is consistently above zero).*

   The text was modified as following:

The measurement data in Fig. 3b and Fig. 3d also demonstrate that for D-glucose and AS/Gl (1:1) particles in the vicinity of RH ≈ 48% the $g_b(RH)$ dependence has an inflection point, (derivative $dg_b/dRH$ has extremum, see insert in Fig. 3b and Fig. 3c) indicating a distinct water sorption mechanism, and hence a different phase state of the particles before and after this RH point.

*Line 205-210: Please provide more details on how the uncertainties are estimated.*

   Line 89, the following sentence has been added: A detailed calculation of growth factor uncertainty is described in Mikhailov et al., 2020 (Section 2.5).

---

## Author Comment (AC2)

Response to referee #2

The referee's comments are in italics, our responses in plain font.
*This work reported water activity and surface tension of aqueous AS and D-glucose aerosol particles using DKA. The manuscript fits well to the scope of ACP. I recommend it to be published after the following comments have been adequately addressed.*

We thank referee #2 for suggestions for improvement that were taken into account upon manuscript revision. Responses to individual comments are given below.

*1. I am worried about the novelty. AS and D-glucose chemicals are not new in hygroscopicity study. Growth factor, water activity and surface tension have been reported in many papers, but not cited in the manuscript. I would suggest the authors demonstrate the new findings (maybe the mixtures and high RH) relative to the previous studies.*

We cannot agree with statement that the manuscript does not include data on previously reported "growth factors, water activity, and surface tension of AS and D-glucose". Sections 3.3 and 4.3 contain references to early modelling and experimental studies. These are used extensively in the following sections when comparing DKA-derived $a_w$ and $\sigma$ for both pure and mixed particles. Overall, 45 literature sources were used. The third and fourth paragraphs of the "Introduction" briefly describes the results of the comparison.

Please note, in contrast to the generally accepted approach, where the measured growth factors are compared with thermodynamic model values. In this paper, the inverse problem is solved, i.e., the thermodynamic parameters of the Koehler equation are determined from the measured dependence of the aerosol growth factor $g_b(RH)$. This is the essence and novelty of the work. As a result, we do not compare the measured $g_b(RH)$ dependences with literature, but rather the thermodynamic parameters derived from them.

*2. Section 4.1: why is the distinct trend of restructuring of AS/Gl particles ($g_{b,H\&D}$) between mass ratio of 4:1 and 1:1, as shown in Figs.1c and 1d.*

The observed differences between AS:Gl particles with mass ratios of 4:1 and 1:1 are due to the different morphology of the dry particles. The 4:1 particles are porous and irregularly shaped, while the 1:1 particles are compact spheres without voids. In the H&D mode, the porous 4:1 particles are restructured by water absorption into compact particles due to the Ostwald-Freundlich effect, while the dry 1:1 particles, already having a compact structure, are not restructured. In the H&D regime, these particles adsorb water on their surface, which is shown by the increase in particle growth ratio with increasing relative humidity (Figure 1c).
Line 99, new clarifying text is added:
The description of the particle restructuring mechanism is beyond the scope of this work. It will be considered in detail in the next paper.

*3. Line 186: any explanation?*
*4. Line 190: is there any evidence about the phase state description?*

Both remarks refer to water uptake by pure Gl and AS:Gl=1:1 at low RH. Lines 185-205 explain why the observed dependence of the growth factor on particle size contradicts the Kelvin effect. Two characteristic features allow us to interpret the phase state of Gl and AS:Gl=1:1 nanoparticles at low RH as a semi-solid amorphous state (Mikhailov et al. 2009; Koop et al., 2011). The first future is the absence of stepwise deliquescence and efflorescence phase transitions, which would be characteristic of crystalline substances. Secondly, the moisture-induced phase transition was found to occur at a "glass transition relative humidity"

of $RH_g \approx 48\%$ (Fig.3b, d). Below the $RH_g$, the particles are mainly in the glassy state. Due to the very low molecular diffusivity of glasses, the uptake of water vapor by glassy aerosol particles is limited to surface adsorption, whereas above the $RH_g$ the particles are in a viscous liquid state and absorb water in the particle bulk. Differences in water sorption mechanisms explain why the growth factors of small particles at low RH are higher than for larger particles.

*5. Figure 9b: could you explain why the surface tension decrease firstly, and then increase along with increasing solution concentration?*

As mentioned in lines 290-304, the decrease in surface tension of AS:Gl = 1:1 aerosol particles is due the fact the $NH_4^+$ and $SO_4^{2-}$ ions effectively neutralize the bipolarity of the monomeric D-glucose molecules, facilitating their association into less polar aggregates with reduced $\sigma$ values, which are more readily accommodated at the air-droplet interface. The subsequent increase of $\sigma$ at high concentrations ($X_s > 0.5$) is the result of particles solidification.

Line 288, new clarifying text is added:

According to Eq. (10) at $X_s \rightarrow 1$, σ approaches the surface tension of a substance in the molten state, $\sigma_s$ which for AS and Gl is 185 mNm$^{-1}$ (Dutcher et al., 2010) and 150.9 mNm$^{-1}$ (Docoslis et al., 2000), respectively. Thus, at $X_s=1$ the mole fraction weighted value of σ for AS:Gl = 4:1 and 1:1 is 179 and 170 mNm$^{-1}$, respectively, which agrees reasonably well with DKA-derived $\sigma$ at high $X_s$ values (Fig. 9a,b).

*6. Is it possible to provide the parametrization for water activity and surface tension of different chemicals using DKA? Then the method could be used more widely.*

The DKA method is used to obtain $a_w$ and $\sigma$ from $g_b(RH)$ dependences. These dependencies can be obtained using various experimental methods, including HTDMA. A feature of DKA is that the $g_b(RH)$ dependence must be obtained for several dry particle diameters in the size range below 100 nm, where the Kelvin term has a significant effect on particle hygroscopicity (Fig. 2). The DKA-derived concentration dependences $a_w$ and $\sigma$ can then be parameterized and used to validate thermodynamic models of nanoparticles. In this paper, some data were parameterized and the resulting coefficients are presented in Tables S1, S2 and S3 in the Supplement.

References

Docoslis, A., Giese, R. F., and van Oss, C. J.: Influence of the water–air interface on the apparent surface tension of aqueous solutions of hydrophilic solutes, Colloids Surf. B, 19, 147-162, https://doi.org/10.1016/S0927-7765(00)00137-5, 2000.

Dutcher, C. S., Wexler, A. S., and Clegg, S. L.: Surface tensions of inorganic multicomponent aqueous electrolyte solutions and melts. The Journal of Physical Chemistry A, 114(46), 12216-12230, https://doi.org/10.1021/jp105191z, 2010.

Mikhailov, E., Vlasenko, S., Martin, S. T., Koop, T., and Pöschl, U.: Amorphous and crystalline aerosol particles interacting with water vapor: conceptual framework and experimental evidence for restructuring, phase transitions and kinetic limitations, Atmos. Chem. Phys., 9, 9491–9522, https://doi.org/10.5194/acp-9-9491-2009, 2009.

Koop, T., Bookhold, J., Shiraiwa, M., and Pöschl, U.: Glass transition and phase state of organic compounds: dependency on molecular properties and implications for secondary organic aerosols in the atmosphere, Phys. Chem. Chem. Phys., 13,19238–19255, https://doi.org/10.1039/C1CP22617G, 2011.